# Therapeutic Opportunities of IL-22 in Non-Alcoholic Fatty Liver Disease: From Molecular Mechanisms to Clinical Applications

**DOI:** 10.3390/biomedicines9121912

**Published:** 2021-12-14

**Authors:** Wenjing Zai, Wei Chen, Hongrui Liu, Dianwen Ju

**Affiliations:** 1Department of Biological Medicines & Shanghai Engineering Research Center of Immunotherapeutics, Fudan University School of Pharmacy, Shanghai 201203, China; 18111010065@fudan.edu.cn (W.Z.); w_chen16@fudan.edu.cn (W.C.); 2Key Laboratory of Medical Molecular Virology (MOE/NHC/CAMS), School of Basic Medical Sciences, Shanghai Medical College, Fudan University, Shanghai 200032, China; 3Multiscale Research Institute of Complex Systems, Fudan University, Shanghai 201203, China; 4Department of Pharmacology, School of Pharmacy, Fudan University, Shanghai 201203, China

**Keywords:** nonalcoholic fatty liver disease, interleukin-22, metabolic syndrome

## Abstract

Nonalcoholic fatty liver disease (NAFLD) represents one of the most common liver disorders and can progress into a series of liver diseases, including nonalcoholic steatohepatitis (NASH), fibrosis, cirrhosis, and even liver cancer. Interleukin-22 (IL-22), a member of the IL-10 family of cytokines, is predominantly produced by lymphocytes but acts exclusively on epithelial cells. IL-22 was proven to favor tissue protection and regeneration in multiple diseases. Emerging evidence suggests that IL-22 plays important protective functions against NAFLD by improving insulin sensitivity, modulating lipid metabolism, relieving oxidative and endoplasmic reticulum (ER) stress, and inhibiting apoptosis. By directly interacting with the heterodimeric IL-10R2 and IL-22R1 receptor complex on hepatocytes, IL-22 activates the Janus kinase 1 (JAK1)/ signal transducer and activator of transcription 3 (STAT3), c-Jun N-terminal kinase (JNK) and extracellular-signal regulated kinase (ERK) pathways to regulate the subsequent expression of genes involved in inflammation, metabolism, tissue repair, and regeneration, thus alleviating hepatitis and steatosis. However, due to the wide biodistribution of the IL-22 receptor and its proinflammatory effects, modifications such as targeted delivery of IL-22 expression and recombinant IL-22 fusion proteins to improve its efficacy while reducing systemic side effects should be taken for further clinical application. In this review, we summarized recent progress in understanding the physiological and pathological importance of the IL-22-IL-22R axis in NAFLD and the mechanisms of IL-22 in the protection of NAFLD and discussed the potential strategies to maneuver this specific cytokine for therapeutic applications for NAFLD.

## 1. Introduction

Nonalcoholic fatty liver disease (NAFLD) is one of the most common causes of liver disorders, ranging from simple steatosis to nonalcoholic steatohepatitis (NASH) (characterized as lobular inflammation and hepatocyte ballooning), with risks of progressive fibrosis, cirrhosis, and even hepatocellular carcinoma (HCC) [1,2,3]. NAFLD is closely associated with obesity, type 2 diabetes mellitus (T2DM), insulin resistance, and atherogenic dyslipidemia, impacting 20% to 30% of adults and 10% of children [1]. NAFLD is also associated with various metabolic syndromes, dysbioses, and gut health [4,5]. However, no therapeutic interventions were approved yet for this disease. Although the pathogenetic drivers and mechanisms of NAFLD were not well recognized, the excessive accumulation of toxic lipid species, which leads to a state of inflammation and provokes endoplasmic reticulum (ER)/oxidative stress, as well as mitochondrial disorders, plays major roles in the development of hepatocyte injury in NAFLD [6,7]. Furthermore, alteration of the intestinal microbiota is also an emerging determinant for NAFLD [8]. Efforts to reverse key causal factors of obesity and fatty liver help relieve hepatocyte injury and improve the liver condition, thereby contributing to the identification of promising treatment candidates. Antioxidants such as vitamin E and the peroxisome proliferators-activated receptor-γ (PPAR-γ) agonist thiazolidinedione were evaluated in clinical trials and shown to reduce hepatic steatosis and relieve inflammation [9,10]. Considering the complex pathogenesis progression of NAFLD, novel targets and therapeutic options are warranted, and the relatively conservative efficacy and potential safety issues of current therapies require evaluation.

Type 3 cytokines, interleukin (IL)-22 and IL-17, which belong to a new class of cytokines that are cosecreted by T helper 17 (Th17) cells, were implicated in multiple aspects of liver injury but have distinct functional spectra [11]. Previous studies indicate that IL-17 mediates liver disease progression by inducing inflammatory tissue responses, while IL-22 is hepatoprotective during liver injury [11,12,13,14]. IL-22 is a member of the IL-10 family of cytokines and is secreted by Th17/22 cells, γδ T cells, natural killer (NK) T cells, and innate lymphoid cells (ILCs) [15]. To the authors’ knowledge, IL-22 is the only cytokine that is produced by immune cells and majorly targets nonhematopoietic cells such as the liver, lung, kidney, pancreas, and intestine [16,17]. Once interacting with the heterodimeric receptor IL-22R complex composed of the subunits IL-22R1 and IL-10R2, IL-22 uses the Janus kinase 1 (Jak1) and tyrosine kinase 2 (Tyk2) cascade to propagate downstream signals and promotes proliferation and tissue regeneration by activating the signal transducer and activator of transcription (STAT) factors STAT1, STAT3 and STAT5 and by interacting with IL-6 and transforming growth factor-β (TGF-β) cascades [17,18]. IL-22 was proven to be protective against liver damage caused by Fas ligand (FASL), carbon tetrachloride (CCl4), concanavalin A (ConA), and alcohol consumption [19,20,21]. Hepatocytes are the major target cells of IL-22, and treatment with exogenous IL-22 not only induces the production of acute-phase proteins (such as α-antichymotrypsin, and serum amyloid A (SAA)) but also promotes hepatic expression of multiple antiapoptotic proteins (such as B-cell lymphoma-2 (Bcl-2) and Bcl-xL), mitogenic proteins (such as Cyclin D1 and cyclin dependent kinase 4 (CDK4)), and antioxidants (metallothionein 1 (MT-1) and MT-2), promoting wound healing, hepatocyte survival and tissue regeneration [22,23]. IL-22 is also described to play roles in host defense within barrier tissues, which protects the epithelial barriers in the gut and mucosa from extracellular pathogens [24]. However, it was also reported that IL-22 is linked to inflammatory tissue pathology and plays roles in the pathogenesis of autoimmune diseases like psoriasis [24]. Considering the many biological functions of this cytokine in disease progression, IL-22 seems to be an attractive target for clinical utility.

Numerous studies indicate that IL-22 can benefit NAFLD in various experimental mouse models [25,26,27]. Recombinant IL-22 (rIL-22) reduces the expression of genes related to lipogenesis and promotes the expression of genes with antiapoptotic, antioxidant and prosurvival activities, thus relieving hepatocyte injury and steatosis, ameliorating insulin resistance, and modulating metabolic syndrome [26]. The efficacy of this cytokine should also be carefully analyzed since contradictory results also exist. The interactions between IL-22 and IL-17 secreted by infiltrating Th cells may contribute to the pathological development of NAFLD, and IL-22 signaling is also suggested to be associated with liver fibrosis [28]. In this review, the biological and pathological functions of the IL-22-IL-22R axis in the progression of NAFLD were summarized and the potential mechanisms of IL-22 in NAFLD treatment were discussed with implications of therapeutically targeting the IL-22-IL-22R axis for further application and modification in the clinic.

## 2. Roles of IL-22-IL-22R Axis in NAFLD

### 2.1. The Role of IL-22 in the Liver

Although comprehensive mechanisms of NAFLD pathogenesis were not yet fully elucidated, recent investigations highlight the importance of the immune response in the progression of NAFLD [29,30]. The increased hepatic infiltration of CD4+ T helper cells and the polarization of the differentiation of Th cells were proven to be essential determinants for the pathological progression of NASH and are closely associated with disease severity (Figure 1) [31]. Evidence shows that Th17 cells increase at the beginning of NASH, leading to fibrosis transition, while Th22 cells peak during the second round of expansion of Th17 cells (both peripherally and in the liver). Th17/22 cells then release cytokines, including IL-6, IL-17, IL-22, tumor necrosis factor-α (TNF-α), TGF-β and C-C motif chemokine ligand 20 (CCL20), in mouse models of NASH and in patients with steatosis, which further promote tissue inflammation and recruitment of leucocytes [28,31,32]. Recent research indicates that NAFLD is also associated with the activation of caspase-1 and IL-1β, as well as increased IL-22 and IL-17A expression [33]. IL-17-deficient (IL-17-/-) mice can be protected from NASH evolution, as indicated by decreased c-Jun N-terminal kinase (JNK) activation, reduced phosphatase and tensin homolog (PTEN) expression (Phosphatidylinositol 3-kinases (PI3K) inhibitor), and improved Akt phosphorylation compared to wild-type mice, accompanied by extensive liver Th22 lymphocyte infiltration. This suggests a possible protective property of IL-22 in NAFLD [19,28]. IL-17 accelerates hepatotoxicity induced by fatty acids (JNK activation) and facilitates the transition from simple steatosis to steatohepatitis/fibrosis and inflammation [28,32,34]. IL-22 exhibits hepatoprotective activity via PTEN stimulation and JNK inhibition, which seems to be an endogenous factor to counteract the effects of IL-17 [28]. Emerging evidence also reveals that the hepatic group 3 ILCs (ILC3s) are increased in high-fat diet (HFD)-fed mice and play vital roles in the protection and pathogenesis of NAFLD by modulating chronic inflammation and liver fibrosis. The protective roles of ILC3s in NAFLD are majorly mediated by IL-22 stimulated by IL-23, which contributes to the upregulation of hepatic lipid metabolism and inhibition of palmitate-induced apoptosis of hepatocytes [35]. The level of secreted IL-22 is also increased in obese leptin-deficient *ob*/*ob* mice treated by IL-12 in combination with IL-18 compared with lean littermates, indicating the potential of fostering tissue repair of this cytokine [36].

### 2.2. The Role of IL-22 in the Gut

The role of gut microbiota was also described in NAFLD, and gut-derived endotoxins act as the “second hits” to promote liver inflammation in the development of NASH, according to the “two hits” model [7,30,37]. Evidence shows that intact mucosal immunity guides diet-induced microbiota dysfunction, thus leading to weight gain in diet-induced obesity (DIO). Circulating lipopolysaccharide (LPS) derived from bacterial products is elevated in obese subjects, suggesting an interplay between metabolic disorders and gut microbiota [4,38]. The lymphotoxin-IL-23-IL-22 pathway, which is essential for gut homeostasis, can regulate the specific commensal response to a high-fat diet. Lymphotoxin β receptor-deficient (Ltbr-/-) mice lack IL-23 and IL-22 and are resistant to diet-induced obesity, indicating that IL-23-IL-22 cytokine signaling regulates the microbiota, thus modulating weight gain and obesity [39]. Restoration of IL-22 expression promotes the production of antimicrobial peptides (including regenerating islet-derived protein 3β (Reg3β) and Reg3γ), which subsequently antagonizes the growth of some microbes, reduces segmented filamentous bacteria (SFB), and normalizes body size [39,40]. The production of antimicrobial peptides by IL-22 therapy has also been reported to maintain gut epithelia and immune barrier integrity against alcohol-induced burn injury [41]. The IL-23-IL-22-microbiota axis is also essential for protecting mice from diet-induced atherosclerosis by controlling intestinal microbial homeostasis [42]. Innate and adaptive lymphocytes successively modulate the gut microbiota and tissue metabolic homeostasis, and IL-22 induces STAT3 phosphorylation in intestinal epithelial cells, which subsequently reduces the expression of lipid transporters in the gut and decreases serum lipid concentration. These results together suggest a role of IL-22 in regulating lipid metabolism [43]. IL-22R1-deficient but not IL-22-deficient mice display higher weight gain and develop higher insulin resistance and glucose intolerance after HFD feeding compared with control mice, suggesting a redundant function of other IL-22R1 ligands, such as IL-20 and IL-24 [27,44]. IL-22 also preserves the endocrine functions and the gut mucosal barrier, alleviates epithelial injury, and restores gut integrity, thus preventing LPS-induced systemic endotoxemia and adipose inflammation during obesity, regulating lipid metabolism in the liver and adipose tissue, and ameliorating hepatic steatosis via STAT3 activation [44]. These results further indicate that IL-22 can be utilized as a novel therapeutic intervention approach in metabolic disorders by regulating gut homeostasis.

### 2.3. The Role of IL-22 in Extrahepatic Comorbidities

Other extrahepatic comorbidities, such as metabolic syndrome, type 2 diabetes mellitus (T2DM) and psoriasis, was also identified to be correlated with the pathophysiology of NAFLD and associated with disease severity [5]. Steatohepatitis promotes the progression of psoriasis in patients with NASH, and obesity exaggerates the severity of psoriasiform dermatitis caused by the Toll-like receptor 7 (TLR-7) agonist imiquimod (IMQ) by promoting IL-17A and IL-22 secretion and IL-1β overexpression in HFD-fed mice [33,45,46]. IL-22/IL-17/IFN-γ-producing T cells are elevated in infiltrated lymphocytes in the pancreas of BDC2.5 T cell receptor transgenic nonobese diabetic (NOD) mice, and the expression level of the IL-22 receptor is high in the pancreas [47]. Increased IL-22 expression does not contribute to the pathogenesis of disease but plays a protective and regenerative role in pancreatic islets by upregulating antiapoptotic factors such as Bcl-2/Bcl-xL, regulating oxidative/ER stress caused by cytokines or glucolipotoxicity, and restoring insulin and glucose homeostasis [44,47]. Endogenous IL-22 content is decreased in diabetic wounds during the inflammatory phase, and IL-22 treatment accelerates the healing process and promotes keratinocyte proliferation by activating the STAT3 cascade [48]. IL-17/IL-22-producing CD4+ T cells also infiltrate the liver and adipose tissue of type 2 diabetic obese patients and are responsible for metabolically abnormal insulin-resistant obesity (MAO). The polarization of CD4+ Th cells is triggered by macrophage-derived IL-1β and can subsequently cause metabolic dysfunction by producing cytokines like IL-6 and TNF-α [49,50]. IL-22 and IL-17 production is suggested to induce inflammatory responses and lead to insulin resistance by activating JNK signaling in insulin-targeting hepatocytes, which was proven to correlate with hepatic lipid accumulation and steatohepatitis [49]. IL-22 conversely promotes pro-IL-1β and IL-1β production in macrophages by activating the c-Jun signaling. Both IL-22 and IL-1β play key roles in paracrine inflammation and participate in the pathogenesis of obese-induced type 2 diabetes. IL-22-associated secretion of IL-1β and IL-10 can also stimulate extracellular-signal regulated kinase (ERK) phosphorylation and promote tumor development in collaboration with diet-induced obesity [51]. Although IL-22 was proven to exert protective responses in many pathological conditions, its extrahepatic complications might also contribute to mediating adipose tissue inflammation and insulin resistance in some circumstances [50].

## 3. Mechanisms of IL-22 on Protection of NAFLD

### 3.1. Hepatoprotecitve Properties of IL-22

The hepatoprotective properties of IL-22 in STAT3-dependent manners were demonstrated in multiple acute and chronic liver diseases [12,52]. Once bound to its heterodimeric receptor complex on the surface of hepatocytes, IL-22 triggers the activation of Jak1-Tyk2 kinases, leading to the phosphorylation of STAT1/3/5 and activation of mitogen activated protein kinase (MAPK) signaling (ERK1/2, MEK1/2, p38 kinase and JNK) (Figure 2) [53]. IL-22 then provokes the expression of its downstream pathway genes, including innate immune mediators, mitogenic modulators (Cyclin D1, CDK4, c-myc, Rb2) and antiapoptotic modulators (myeloid cell leukemia-1 (Mcl-1), Bcl-2, Bcl-xL), thus protecting hepatocytes from damage and promoting survival, proliferation, and liver regeneration [16,20]. IL-22 is reported to activate various mitochondrial DNA repair genes (such as endonuclease-VIII-like-1 (Neil-1) and 8-oxoguanine DNA glycosylase (OGG-1)), antioxidant genes (such as MT-1 and MT-2) and acute phase genes (such as SAA) to protect hepatic cells [54]. The positive correlations between IL-22 secretion and proliferating-cell nuclear antigen (PCNA) expression suggest that IL-22 mediates liver regeneration during recovery of drug-induced liver injury (DILI) [55]. Administration of IL-22 protects mice from alcohol-induced liver injury and fatty liver, as evidenced by reduced lipid peroxidation and restored glutathione (GSH) levels, by STAT3-dependent activation of antioxidant, antiapoptotic and antimicrobial genes, and by restoring the balance of reactive oxygen species (ROS) and inhibiting TNF-α-mediated hepatocyte apoptosis [21,56,57]. The hepatoprotective properties of IL-22 are also observed in NAFLD and are majorly mediated by stimulating STAT3 phosphorylation, increasing PTEN and preventing PI3K/Akt activation, followed by modulating downstream gene expression, including antiapoptotic, antioxidative and lipogenic genes in hepatocytes, while evidence indicates that it is effective only in the absence of IL-17 [28,34,58]. IL-22 was also reported to be involved in the therapeutic effects of blueberry on NAFLD by inhibiting apoptosis through the JAK1/STAT3/BAX signaling pathway [59].

### 3.2. Metabolic Reprograming Effects of IL-22

Evidence shows that IL-22 can directly modulate lipid metabolism by promoting lipolysis and enhancing fatty acid β-oxidation in adipose tissue and hepatocytes of obese mice, thus attenuating obesity-associated fatty liver and steatosis [27,52]. The action of IL-22 in regulating metabolic homeostasis was described, since treatment of *db*/*db* mice or mice fed a HFD with rIL-22 strongly reduces body weight, decreases blood glucose levels, and reverses various metabolic symptoms, including insulin resistance and hyperglycemia [27]. Low-grade chronic intestinal inflammation, which is associated with HFD-induced oxidative/ER stress, was observed in HFD-induced obese mice. IL-22 can significantly reduce HFD-induced oxidative/ER stress and inflammation and reverse microbial changes associated with obesity [60]. Moreover, treatment with recombinant murine IL-22 (rmIL-22) was proven to decrease lipid content and hepatic steatosis in *ob/ob* mice fed a HFD via reducing the expression of genes associated with lipogenesis (such as sterol regulatory element-binding protein-1c (SREBP-1c), fatty acid synthase (FAS), ATP citrate lyase (ACLY) and elongation of very-long-chain fatty acids protein 6 (ELOVL6)), and reduces expression of gluconeogenesis-related enzymes (such as phosphoenolpyruvate carboxykinase (PEPCK)) in murine [26,27]. IL-22 also acts on adipocytes and modulates genes related to triglyceride lipolysis (such as patatin-like phospholipase domain-containing protein 2 (PNPLA2)) and fatty acid β-oxidation (such as acyl-CoA oxidase 1 (ACOX1)) [27]. Other studies indicate that IL-22 drives metabolic adaptive reprogramming by upregulating adenosine monophosphate (AMP)-activated protein kinase (AMPK), mammalian target of rapamycin (mTOR) and Akt signaling in STAT3-dependent manners and promotes mitochondrial oxidative phosphorylation (OXPHOS) and glycolysis to maintain mitochondrial fitness in hepatocytes, thus protecting against liver injury and HFD-induced steatohepatitis [61].

### 3.3. Microbiota Protective Functions of IL-22

Considerable evidence indicates that dysbiosis accelerates the pathogenetic progression of NAFLD by increasing hepatic exposure to injurious substances of bacterial products that permeate from the gut [4]. Insulin restores gut health and microbiota loads as well as IL-22 production, prevents microbiota encroachment, and protects against HFD-induced low-grade inflammation (LGI) and metabolic syndrome in an IL-22-dependent microbiota manner. Mechanistically, IL-22 protects against HFD-mediated inflammatory challenges and metabolic syndrome by promoting epithelium metabolism and proliferation, thus shielding protection against the gut microbiota [62]. The gut microbiota promotes IL-22 production in pancreatic ILCs, and IL-22 modulates β cell function and regulates inflammation by stimulating the expression of mouse β-defensin 14 (mBD14) in pancreatic β cells of NOD mice [63]. Microbiota-dependent production of IL-22 by rewiring tryptophan metabolism with indoleamine 2,3-dioxygenase inhibitor preserves the gut mucosal barrier, improves insulin sensitivity, modulates lipid metabolism, and decreases endotoxemia and chronic inflammation in the liver and adipose tissue [64]. Increased IL-22 production by intestinal ILC3s also participates in improved polycystic ovary syndrome (PCOS) and relevant insulin resistance [65]. Downstream activities of IL-22, such as maintaining mucosal epithelial integrity, alleviating metabolic dysfunctions, and reducing inflammation, may all contribute to the protective functions of IL-22 on NAFLD, obesity, and other metabolic syndromes [66]. However, evidence shows that endogenous IL-22 and biologically active IL-22 do not influence the development of adiposity or metabolic alterations, nor do they influence body weight, insulin resistance, or glucose tolerance in mice. The lack of phenotypes can be explained by relatively low levels of inflammation and low numbers of circulating IL-22-producing cells. Pharmacological treatment with exogenous rmIL-22 fusion protein downregulates the expression of hepatic glucogenesis-related genes and reduces blood glucose levels via activation of STAT3 and AMPK signaling in hepatocytes [67]. These results indicate that IL-22 can be used as a potential therapeutic candidate for metabolic syndromes such as NAFLD, as well as other extrahepatic comorbidities such as autoimmune diabetes and type 1 diabetes (T1D) [68].

## 4. Concerns of IL-22 in NAFLD Application

IL-22 possesses both tissue-protective effects and immunomodulatory properties, inducing not only mitogenic and antiapoptotic genes in hepatocytes but also pro- and anti-inflammatory genes [17,69]. The functional consequence of IL-22 in immune regulation can be either pathologic or protective, making it a highly controversial cytokine [70,71,72]. By upregulating the expression of chemokines including chemokine (C-X-C motif) ligands CXCL1/5/9, and cytokines like IL-6 and granulocyte colony-stimulating factor (G-CSF), administration of exogenous IL-22 is sufficient to trigger systemic or local inflammation [17]. The most prevalent view is that IL-22 possesses protective roles in liver injury during hepatitis, which relieves tissue damage by counteracting the destructive nature of immune responses [13]. Conversely, alternative insights also arise, indicating that IL-22 may participate in the progression of disease. These controversial actions of IL-22 might be dependent on the concentration, exposure time and tissue microenvironment [55]. For example, IL-22 plays a pro-inflammatory pathological role to mediate the inflammatory response in the liver of hepatitis B virus (HBV) transgenic (TG) mice after HBV-specific T cell recognition [69]. Positive correlations between liver-infiltrating IL-22+ cells and liver fibrosis stage were identified, and blockade of IL-22 significantly reduces hepatic lymphocyte recruitment and fibrosis progression by relieving the expression of CXCL10 and CCL20 in HBV Tg mice [73]. The coexistence of IL-17A with IL-22 augments the proinflammatory effects of IL-22, which might worsen the disease situation [73,74]. Pathogenic roles of the IL-22 and IL-17-mediated type 3 immune response are identified in the development of liver fibrosis in MAPK-dependent manners by enhancing TGF-β signaling in hepatic stellate cells (HSCs) [72]. IL-22RA1 knockout mice or mice with blockade of IL-22/IL-17 production by RAR-related orphan receptor gamma-t (RORγt) or aryl hydrocarbon receptor (AhR) antagonists exhibit reduced liver fibrosis [72]. IL-22 is also associated with tumorigenesis, promoting proliferation of transformed cells [12,55]. Liver-specific IL-22 transgenic (IL-22TG) mice are resistant to concanavalin A-induced liver injury but are more susceptible to tumor development with limited influence on liver inflammation [75]. A soluble form of the IL-22R homolog, IL-22 binding protein (IL-22BP), is observed to bind with IL-22 and regulate IL-22-related pathways. It was also reported to counterbalance the proinflammatory effects of IL-22, while the biological functions and regulation of the balance of endogenous IL-22/IL-22BP ratio are still unknown [76]. These complicated, sometimes opposite, biological effects account for the difficulties of its clinical application.

## 5. Therapeutic Applications of IL-22 for NAFLD

The lack of therapeutic treatment is a central challenge for NAFLD management considering the complex disease pathologies, including steatosis, inflammation, fibrosis, and hepatocyte apoptosis. Modulation of the IL-22-IL-22R axis seems to be an ideal therapeutic strategy for NAFLD, considering its multiple functions, such as antioxidative, antiapoptotic and proregenerative effects, by activating STAT3 cascades in hepatocytes (Table 1). IL-22 also possesses limited side effects and does not influence the immune system due to the restricted biodistribution of IL-22R1 to epithelial cells [76,77]. IL-22TG mice with high serum levels of IL-22 (~6,000 pg/mL) exhibit normal compared to wild-type mice, except for a slower body weight gain and minor skin inflammation [75]. Considering the rapid clearance rate of IL-22 in vivo, strategies to prolong the half-life of this cytokine, such as fused expression with the Fc proportion of human immunoglobulin (Ig), are necessary for its clinical utilization [78]. There are currently two rIL-22-based therapies, including F-652 (IL-22 fused with constant region of IgG2) and UTTR1147A (IL-22 fused with IgG4), in clinical trials exploring their tissue protective functions in several inflammatory diseases (alcoholic hepatitis (AH), inflammatory bowel disease (IBD), acute graft-versus-host disease (GvHD) and neuropathic diabetic foot ulcers) [79]. The first-in-human phase I clinical study of F-652 shows that IL-22Fc is well tolerated with favorable pharmacological (PK) and pharmacodynamic (PD) properties in healthy humans, with no severe adverse events observed but skin reaction symptoms due to IL-22-mediated STAT3 activation [80]. F-652 is biologically active, as evidenced by increased SAA and C-reactive protein (CRP) levels and decreased hepatic expression of enzymes associated with lipogenesis. No evidence of a systemic inflammatory response is identified other than elevation of the acute phase response [80]. A phase I placebo-controlled trial of single, ascending doses of UTTR1147A in healthy volunteers confirmed the safety and tolerability of this fusion protein, and the results showed that prolonged exposure to IL-22Fc fusion proteins with a half-life of ~1 week efficiently induced marker gene expression (such as Reg3a, SAA and CRP) without triggering inflammatory symptoms [81]. F-652 was also assessed in a phase II dose-escalating study for its safety and efficacy in patients with alcoholic hepatitis. No serious adverse events were reported, and distinct improvement was determined, as evidenced by improved Lille and Model for End-Stage Liver Disease [MELD] scores, reduced inflammation markers and increased hepatic regeneration rates [57,82]. Cytokine analysis supports the idea that IL-22 is tissue protective but anti-inflammatory, as it upregulates regeneration proteins such as fibroblast growth factor-β (FGF-β), platelet-derived growth factor-AA (PDGF-AA), and PDGF-BB and downregulates inflammatory markers, including IL-8, IL-6, CRP, and monocyte chemoattractant protein-1 (MCP-1) [82]. The therapeutic potential and liver regeneration properties of F-652 were further evaluated in acute-on-chronic liver failure (ACLF). Administration of IL-22Fc significantly improves the survival of ACLF mice, reverses STAT1/STAT3 pathway imbalance, and attenuates bacterial infection by reprogramming impaired regeneration signaling [83]. Since IL-22 does not influence immune cells, administration of rhIL-22 to patients with pancreatic injury or GvHD relieves the disease state and supports protection and tissue regeneration without triggering immune-related side effects [22]. These tissue protective and regenerative functions of IL-22 together suggest the therapeutic potential of this cytokine for the treatment of liver diseases [22,84].

Similarly, IL-22Fc relieves CXCL1-driven NASH in methionine-choline deficient diet (MCD)- or HFD- fed mice via STAT3-dependent activation of antioxidant enzyme (MT-1 and MT-2) expression, which eradicates superoxide and hydroxyl radicals in the liver [76]. Administration of rmIL-22 or IL-22 adenovirus activates hepatic STAT3 signaling, therefore alleviating alcohol-induced hepatitis and steatosis. IL-22 treatment also downregulates fatty acid transport gene expression while upregulating antioxidant, antiapoptotic and antimicrobial genes in mice [21]. Other therapeutic agents that buster IL-22 secretion or activate IL-22 signaling may also represent attractive strategies. Resiquimod (R848), a synthetic TLR-7 ligand that stimulates innate antimicrobial immune defenses, was proven to induce IL-23 expression and subsequently trigger IL-22 production by interacting with CD11c+ dendritic cells [85]. IL-22 also acts as a crucial mediator of the hepatoprotective effects of TLR-7 agonistic agent (1Z1) and functions by promoting the expression of antimicrobial peptides (Reg3b and Reg3g) and modulating the intestinal microbiome in an alcoholic hepatitis mouse model [86]. The activation of TLR-7 has also been proven to prevent liver fibrosis [87]. Increased NKG2A expression by NK1.1+ cells in a T1D mouse model can enhance allograft survival, increase insulin secretion, and reduce the inflammatory response to allografts in an IL-22-dependent manner. Vaccination with a TLR-9 agonist (ODN1585, a CpG oligonucleotide) enhances hepatic IL-22-producing CD3-NK1.1+ cell expansion, thus prolonging the survival of pancreatic islet allograft in the liver parenchyma of T1D mice and increasing insulin secretion [88]. Metabolic syndrome is reported to be associated with impaired AhR agonist production by gut microbiota-dependent tryptophan metabolism, and treatment with an AhR agonist (Ficz, 6-formylindolo (3,2-b) carbazole) or engineered *Lactobacillus reuteri* strain with superior tryptophan metabolizing capacities can relieve metabolic disorders and liver steatosis in HFD-fed mouse models, primarily by inducing intestinal IL-22 production and cytochrome P450 1A1 (CYP1A1) gene expression [89]. Administration of an AhR agonist to mice has also been reported to improve alcohol-induced liver injury and can defend against gut inflammation and relieve hepatic steatosis in obesity and NAFLD [90,91,92]. Applying engineered *Lactobacillus reuteri* to produce IL-22 or restoring intestinal levels of indole-3-acetic acid (IAA) to induce IL-22 expression to mice fed an ethanol diet relieves ethanol-induced liver damage and inflammation [93]. Production of rIL-22 by engineered *Lactobacillus reuteri* can also ameliorate hepatic steatosis in mice with diet-induced obesity, as evidenced by reduced triglyceride levels and liver weight via the IL-22/STAT3/Reg3 cascade [94]. The roles of IL-22 in metabolic regulation opens new avenues for its potential application in NAFLD and other metabolic disorders [95].

Considering the plenty of effects of IL-22, the clinical application of this cytokine should be carefully evaluated. Targeted delivery of IL-22-expressing genes or IL-22 fusion proteins to the liver might enhance its hepatoprotective effects while reducing its systemic side effects. The very first attempt was reported in 2004, in which hydrodynamic injection of an IL-22 expression vector with a liver-specific albumin promoter by the tail vein achieved efficient IL-22 expression, STAT3 activation and liver protection [96]. An IL-22-Ig fusion gene vector was constructed, evaluated in rat experimental autoimmune myocarditis (EAM) by hydrodynamic-based gene delivery, and exhibited tissue protection [97]. Antibody-based fusion protein, comprising murine IL-22 fused at the N-terminus of F8 antibody specific to extradomain A (IL-22-F8), in diabody format, was analyzed in dextran sodium sulfate (DSS)-induced colitis in mice. This fusion protein selectively accumulated at the site of disease and showed rapid improvement of disease syndrome [98]. Gene therapy of IL-22 fused with apolipoprotein A-1 (ApoA1) achieved sufficient liver-targeted cytokine expression and protected mice from acetaminophen-induced liver damage without initiating an inflammatory response or systemic toxicity [99]. Liver-targeted delivery of the IL-22-ApoA1 fusion gene by penetratin-based hybrid nanoparticles relieved liver injury, evaded immune responses, promoted hepatocyte regeneration, and relieved oxidative stress and mitochondrial dysfunction in a concanavalin A-induced hepatitis mouse model [100]. Further targeted IL-22-ApoA1 gene delivery in the liver significantly alleviated hepatitis steatosis and improved metabolic syndromes such as insulin resistance by modulating the STAT3/ERK1/2 and nuclear factor erythroid 2-related factor 2 (Nrf2)/superoxide dismutase 1 (SOD1) signaling, and lipid metabolic gene expression in an HFD-fed mouse model [25]. These targeted nanoparticles and fusion genes can be exploited as novel, safe and efficient strategies for the management of NAFLD. Combinational therapies have also been exploited. A novel vascular endothelial growth factor B antibody (anti-VEGFB)/IL-22 fusion protein reduced renal and hepatic lipid accumulation and insulin resistance by downregulating fatty acid transporting gene expression and regulating glycolipid metabolism and ameliorated inflammatory responses by relieving oxidative stress and mitochondrial dysfunction, thus protecting *db/m* and HFD-fed *db/db* mice from diabetic nephropathy and liver steatosis [101]. Vunakizumab-IL-22 fusion protein (vmab-IL-22), composing IL-22 fused to the C-terminus of the anti-IL-17A antibody, exhibited favorable biological activities in promoting tissue protection and inhibiting excessive inflammation in influenza A virus-induced lung injury in mice [102]. Further exploration of novel therapeutic strategies using bioengineering and pharmacological technologies and evaluation of their biosafety and biological activities are necessary for clinical applications.

**Table 1 biomedicines-09-01912-t001:** Therapeutic applications of IL-22 for human diseases.

Therapeutics	Description	Indication	Reference
F-652	hIL-22 and IgG2 fusion protein	Alcoholic hepatitis; GvHD	[79,80,82]
UTTR1147A	hIL-22 and IgG4 fusion protein	IBD; neuropathic diabetic foot ulcers	[79,81]
IL-22Fc	mIL-22 and IgG fusion protein	ACLF; NAFLD; alcoholic hepatitis	[21,27,76,83]
R848	TLR-7 agonist, IL-22-inducing agent	Vancomycin-resistant enterococcus	[85]
1Z1	TLR-7 agonist, IL-22-inducing agent	Alcoholic hepatitis	[86]
ODN1585	TLR-9 agonist, IL-22-inducing agent	Type 1 diabetes	[88]
Ficz	AhR agonist, IL-22-inducing agent	Alcoholic hepatitis	[89,90]
*Lactobacillus reuteri*	Engineered bacteria that produce IL-22	Alcoholic hepatitis; NAFLD	[93,94]
IL-22-ApoA1	mIL-22 fused to ApoA1 gene therapy vector	Liver injury; NAFLD	[25,99,100]
IL-22-F8	mIL-22 and F8 antibody fusion protein	Colitis	[98]
Anti-VEGFB/IL-22	mIL-22 and anti-VEGFB fusion protein	Diabetic nephropathy; fatty liver	[101]
Vmab-IL-22	mIL-22 and anti-IL-17A antibody fusion protein	Influenza A virus infection	[102]

Abbreviations: TLR: Toll-like receptor; GvHD: acute graft-versus-host disease; IBD: inflammatory bowel disease; ACLF: acute-on-chronic liver failure; NAFLD, nonalcoholic fatty liver disease; VEGFB, vascular endothelial growth factor B.

## 6. Conclusions

An increasing number of studies based on clinical observations and experimental mouse models highlight the opportunities for targeting the IL-22-IL-22R axis as novel, promising therapeutic strategies for NAFLD. The expression levels of IL-22 and IL-17 are increased during the pathological progression of NASH. IL-17 accelerates disease severity, facilitating liver inflammation and steatohepatitis/fibrosis transition. IL-22 exhibits hepatoprotective properties by inducing antioxidant, antiapoptotic, prosurvival, and proliferative genes in a STAT3-dependent fashion [32,33,34]. Administration of recombinant IL-22 fusion protein or induction of IL-22 signaling by therapeutic agents was proven to relieve hepatic steatosis and reverse various metabolic symptoms by downregulating expression of genes related to lipogenesis and triglyceride synthesis, alleviating lipotoxic substrate-induced oxidative/ER stresses, and maintaining intestinal barrier integrity in *db/db*- or HFD-fed obese mice [26,27]. Taken together, these results suggest that IL-22 and IL-22-related signaling appear to be promising therapeutic targets for NAFLD, although stringent investigations are warranted in the future. More careful investigation of the biological and pathological functions of IL-22 in disease progression, an in-depth understanding of the interactions of IL-22 with other cytokines like IL-17 and the endogenous neutralizer IL-22BP, and further exploration of the internal mechanisms by which IL-22 protects against NAFLD are warranted. In addition, bioengineering and pharmacological technologies might help improve its internal circulating time and liver biodistribution, reduce its side effects, and improve its protective properties. The biological efficiency and tissue safety of these therapeutics should also be carefully evaluated for their clinical application.

The present review indicates that the IL-22-IL-22R axis plays significant physiological and pathological functions in the progression of NAFLD, and pharmacological treatment with exogenous IL-22 or IL-22-producing agents can relieve disease severity via multiple mechanisms, suggesting that IL-22 can be utilized as a novel target for NAFLD intervention. However, more efforts should be made to maneuver this specific cytokine for therapeutic applications.

## Figures and Tables

**Figure 1 biomedicines-09-01912-f001:**
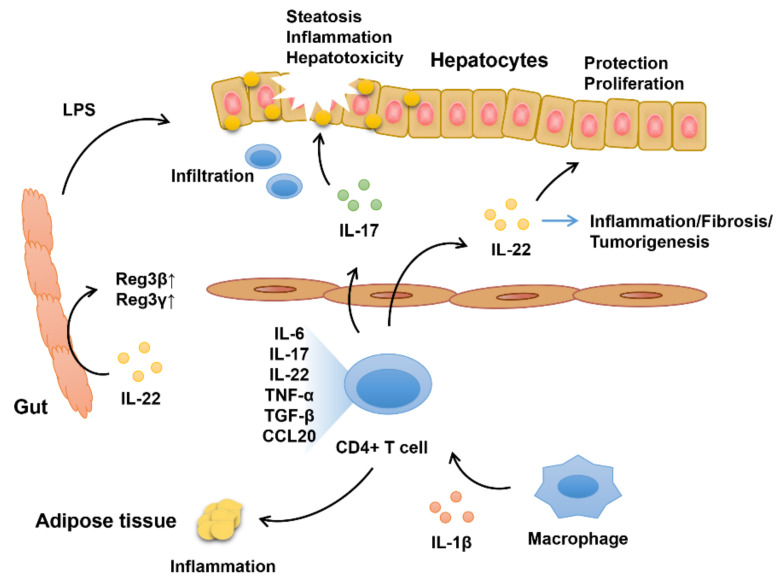
Schematics of roles of IL-22 on NAFLD. Expression of levels of IL-22 and IL-17 that secreted by infiltrating CD4+ T-helper cells is elevated. IL-17 accelerates inflammation and disease severity, whereas IL-22 exhibits tissue protection and proproliferation properties. However, concerns should also be taken since IL-22 might also participate in inflammation, liver fibrosis, and tumorigenesis.

**Figure 2 biomedicines-09-01912-f002:**
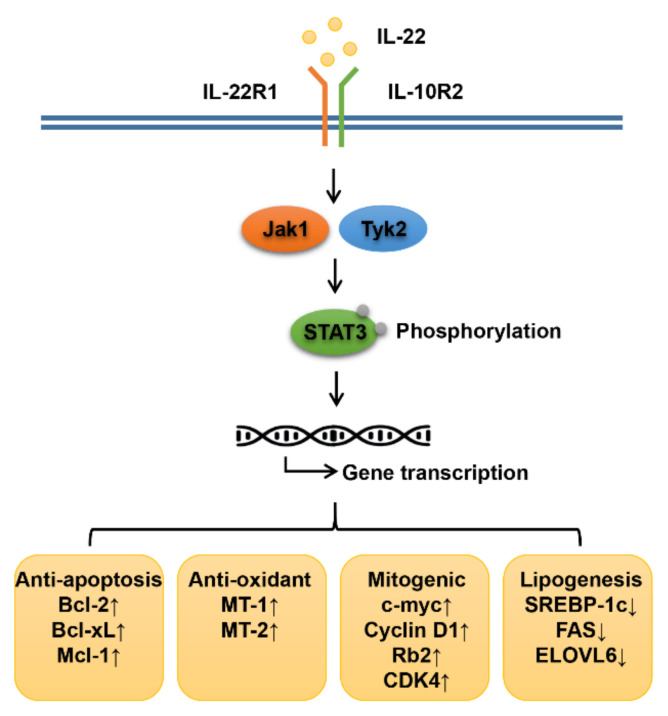
Protective mechanisms of IL-22 on NAFLD. Once interacting with its heterodimeric receptor complex IL-22R1 and IL-10R2, IL-22 induces activation of Jak1-Tyk2 kinases and STAT3 phosphorylation. IL-22 then provokes multiple downstream gene expression, including antiapoptotic, antioxidant and mitogenic genes. IL-22 also modulates lipid metabolism by enhancing lipolysis while relieving lipogenesis in STAT3-dependent fashions.

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
