# Peer review of "Therapeutic Opportunities of IL-22 in Non-Alcoholic Fatty Liver Disease: From Molecular Mechanisms to Clinical Applications"

_biomedicines, 2021, doi:10.3390/biomedicines9121912_

Round 1

Reviewer 1 Report

Title: Therapeutic Opportunities of IL-22 in Non-Alcoholic Fatty Liver Disease: From Molecular Mechanisms to Clinical Applications

Authors: Wenjing Zai, Wei Chen, Hongrui Liu, Dianwen Ju

General Comment:

Non-Alcoholic Fatty Liver Disease (NAFLD) is nowadays a leading cause of liver failure and indication for liver transplantation; however, the therapeutic options for this treatment are limited. Therefore, there is a constant need for novel NAFLD treatments. In their work, Wenjing Zai et al. reviewed the present concepts on the role of interleukin 22 in NAFLD development and discussed the potential strategies targeting this cytokine for therapeutic applications for NAFLD. The manuscript is an example of a solid review of the literature; therefore, I have only some minor remarks regarding its structure that the Authors should address before it is accepted for publication.

  • Sections: 2 (Roles of IL-22-IL-22R axis in NAFLD) and 3 (Mechanisms of IL-22 on protection of NAFLD)

Please consider dividing these sections into subsections referring to distinct potential molecular mechanisms of IL-22 action.

  • Section 2. Roles of IL-22-IL-22R axis in NAFLD

“Other extrahepatic complications in NAFLD have been reported, such as metabolic syndrome, type 2 diabetes mellitus (T2DM) and psoriasis.”  - this sentence is “unfortunate" from the clinical point of view, NAFLD is considered as the sixth component of the metabolic syndrome or its hepatic manifestation. In turn, hepatic insulin resistance is only one of the elements of the insulin resistance observed in T2DM patients.

  • Section 3: 3. Mechanisms of IL-22 on protection of NAFLD

“IL-22 therapy decreases hepatic expression of lipid synthesis-related genes and reduces expression of gluconeogenesis-related enzymes (such as ACLY and PEPCK).” – please provide proper reference and explain if this finding refers to an animal or human studies

  • Section 4. Therapeutic applications of IL-22 for NAFLD

“A phase I placebo-controlled trial of UTTR1147A confirmed that prolonged exposure to IL-22 fusion proteins with a half-life of ~1 week efficiently induced marker gene expression without triggering inflammatory symptoms [86].” – please provide some details of the study design.

“IL-22TG mice with high serum levels of IL-22 (~6000 pg/ml) exhibits” – please consider changing to exhibit.

  • Section 5. Conclusions

“In conclusion, the present study indicates…” – please consider replacing “study” by review.

  • Abstract:

„Interluekin-22” – please change to ”Interleukin-22”

  • Whole manuscript:
  • please explain the abbreviations as they occur in the text, e.g., TNF-α, TGF-β, CCL20, PCNA, DILI, T1D, etc.
  • please use a consistent punctation e.g. “and alcohol consumption[19-21].” or “and recruitment of leucocytes [28, 31, 32]." 

Reviewer 2 Report

This review is informative and well organized. I have no further major comment, but I recommend to add recent article regarding about this review to enrich the content.

Hamaguchi et al., Group 3 Innate Lymphoid Cells Protect Steatohepatitis From High-Fat Diet Induced Toxicity Front. Immunol. 12:648754 (2021) doi: 10.3389/fimmu.2021.648754

In addition, the abbreviated words should be shown by the full-spelling, when they were first appeared.
